

# 2D hexagonal boron nitride for solar energy conversions

Amall Ahmed Ramanathan

The University of Jordan, Amman, Jordan

## ABSTRACT

The optoelectronic properties of free standing monolayer (ML) hexagonal boron nitride (h-BN) is investigated for potential solar energy conversion applications using the density functional theory (DFT) full potential linearized augmented plane wave (FP-LAPW) method. In addition, the bulk optical properties have also been calculated for the sake of comparison. The dielectric functions, optical conductivities and the optical constants are evaluated using the relaxed structures from electronic total energy pseudopotential calculations. The results reinforce previous research on h-BN DUV optoelectronics and demonstrate the suitability of its use as a component in deep ultraviolet (DUV) and energy conversion devices.

## INTRODUCTION

The last two decades have seen tremendous research on two-dimensional (2D) materials similar to graphene; which was the first 2D material discovered with a honeycomb structure consisting of single layers of carbon atoms and endowed with remarkable electronic, mechanical and optical properties (*Lemme, 2010*; *Garnica, Knaust & Fatikow, 2019*; *Aqra & Ramanathan, 2020*; *Szunerits & Boukherroub, 2018*). Especially, layered transition metal dichalcogenides (TMDs) are widely investigated due to the rich diversity of properties which can be attributed to the finite and tuneable bandgap in these materials and have seen a multitude of applications (*Ramanathan & Khalifeh, 2018*; *Chen et al., 2015*; *Vikraman et al., 2017*; *Ramanathan & Khalifeh, 2021a*).

In comparison to TMDs a less researched graphene analog is h-BN that is now gaining popularity as the key building block in van der Waal hetrostructures (*Geim & Grigorieva, 2013*). Each $B^{3+}$ atom is bonded in a trigonal planar geometry to three equivalent $N^{3-}$ atoms and vice versa as shown in Fig. 1. Similar to graphene, 2D h-BN has a honeycomb structure and consists of alternating B and N atoms that are $sp^2$ covalently bonded, but it has an insulating character. Moreover, due to the difference in electronegativity of the B and N atoms, the bond between B and N shows an ionic character and it is not purely covalent as in graphene. Being the structural analog of graphene h-BN is expected to possess and it does show excellent mechanical properties and high chemical and thermal stabilities (*Zhi et al., 2009*; *Grosjean et al., 2016*; *Ramanathan, Aqra & Al-Rawajfeh, 2018*). The small mismatch in lattice parameter of h-BN with graphene and the absence of dangling bonds and charge traps makes it an ideal substrate for graphene based devices

Corresponding author
Amall Ahmed Ramanathan,
amallahmad@gmail.com

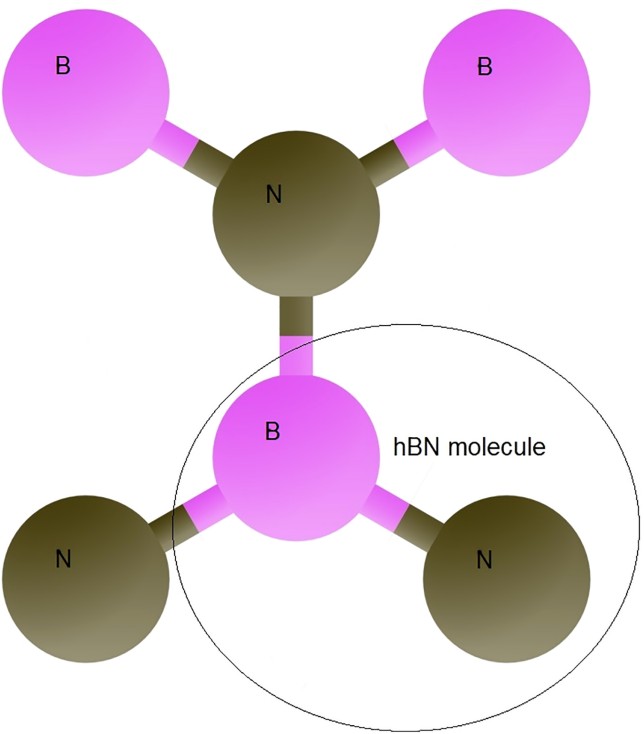

**Figure 1  Illustration of an h-BN molecule.**

(*Wang et al., 2013*; *Khan et al., 2018*). The indirect to direct bandgap crossover in MLs of TMDs with its efficient light matter coupling is also observed in h-BN (*Cassabois, Valvin & Gil, 2016*; *Elias et al., 2019*) and makes it a promising material for optoelectronics and deep ultraviolet (DUV) applications (*Watanabe, Taniguchi & Kanda, 2004*; *Liu et al., 2018*; *Yin et al., 2016*).

It is now the age of nano electronics and with the increasingly small size of components there is a need to extend optical systems to shorter wavelengths than visible light to improve the image resolution. This has motivated the interest in the short-wave UV region, especially DUV optics. The focus of the present study is to investigate the dynamic electrical conductivity and optical constants in order to better understand and substantiate previous research on h-BN DUV optoelectronics and its possible use as a component in solar energy conversion devices.

## METHOD

The calculations are carried out in two steps. First, DFT calculations using the ABINIT software program (*Gonze et al., 2016*) with the generalized gradient approximation (GGA) projector augmented wave (PAW) pseudopotentials (*Perdew, Burke & Ernzerhof, 1996*; *Torrent et al., 2008*) are performed for structural optimization as in previous works (*Ramanathan, 2013*; *Ramanathan & Khalifeh, 2017*). Convergence criteria of less than $1 \times 10^{-6}$ Ha for the self consistent field (SCF) iterations and force threshold of less than 1 mRy/ a.u. for structural optimizations are used with the optimized k-point grids of $10 \times 10 \times 10$

**Table 1 Structural and electronic properties of the relaxed h-BN.**

| Compound | Space group | Lattice constant (Å) | | Bandgap (eV) | | Inter layer spacing | Other works | | |
|---|---|---|---|---|---|---|---|---|---|
| | | a | c | | | | a | c | Bandgap |
| h-BN bulk | P63/mmc #194 | 2.517 | 6.706 | 4.23 | K–M indirect | 3.353 | 2.505[a] 2.511[b] | 6.661[a] 6.688[b] | 5.955[c] |
| h-BN ML | P-6 m 2 #187 | 2.515 | 15.063 | 4.65 | K–K direct | – | | | 6.1[d] |

Notes:
[a] *Lynch & Deickamer (1966).*
[b] *Xu et al. (2015).*
[c] *Elias et al. (2019), Vuong et al. (2017), Schuster et al. (2018).*
[d] *Elias et al. (2019).*

and $10 \times 10 \times 5$ for h-BN bulk and ML h-BN respectively and a cutoff energy of 20 Ha to obtain the lattice constants and relaxed structures.

Next, the relaxed geometries that have been obtained are used to perform FP-LAPW calculations with the WIEN2k (*Blaha et al., 2020*) code employing GGA_PBE to obtain the ground state energies and band structures. The wave functions in interstitial region were expanded in plane waves with a cutoff of Rmt*Kmax set to seven for energy eigenvalues convergence; Kmax gives the magnitude of the largest K vector in the plane-wave expansion and the muffin-tin radii (Rmt) are set to 1.37 and 1.36 a.u for both the atoms in the bulk and ML h-BN respectively. K-point grids of $10 \times 10 \times 10$ and $15 \times 15 \times 2$ are used for the electronic calculations of the bulk and ML respectively. More stringent calculations using denser grids of $19 \times 19 \times 6$ and $50 \times 50 \times 2$ are used for the bulk and ML h-BN respectively to evaluate the optical properties, namely the frequency dependent conductivities, real and imaginary parts of dielectric tensor, index of refraction, reflectivity and the absorption and extinction coefficients.

## RESULTS

### Structural and electronic

The simulation of the 2D BN in the hexagonal structure is done by using the supercell technique. The optimized value of the bulk lattice constant is used to set up the super cell that contains a single atomic layer of h-BN separated by sufficient amount of vacuum in the z direction to offset any interactions between layers. The free standing h-BN ML is relaxed to obtain the ground state. The bulk and ML structures of h-BN are shown in Fig. S1. The lattice constant values and band gaps for the optimized bulk and ML are listed in Table 1 and are consistent with those of other experimental and GGA-PBE calculations (*Elias et al., 2019*; *Lynch & Deickamer, 1966*; *Xu et al., 2015*; *Vuong et al., 2017*; *Schuster et al., 2018*). The band structures of the bulk and ML h-BN are presented in Fig. S2. We see an indirect K–M (bulk) to direct K–K band (ML) transition, consistent with recent results showing a direct wide bandgap value of 4.65 eV in the ML. This is also in accordance with other 2D materials like $MoS_2$ which show and indirect to direct band cross over when going from the bulk to the ML (*Ramanathan & Khalifeh, 2021a*; *Mak et al., 2010*). In the past there was a controversy about the nature of the bandgap of h-BN bulk and ML, but,

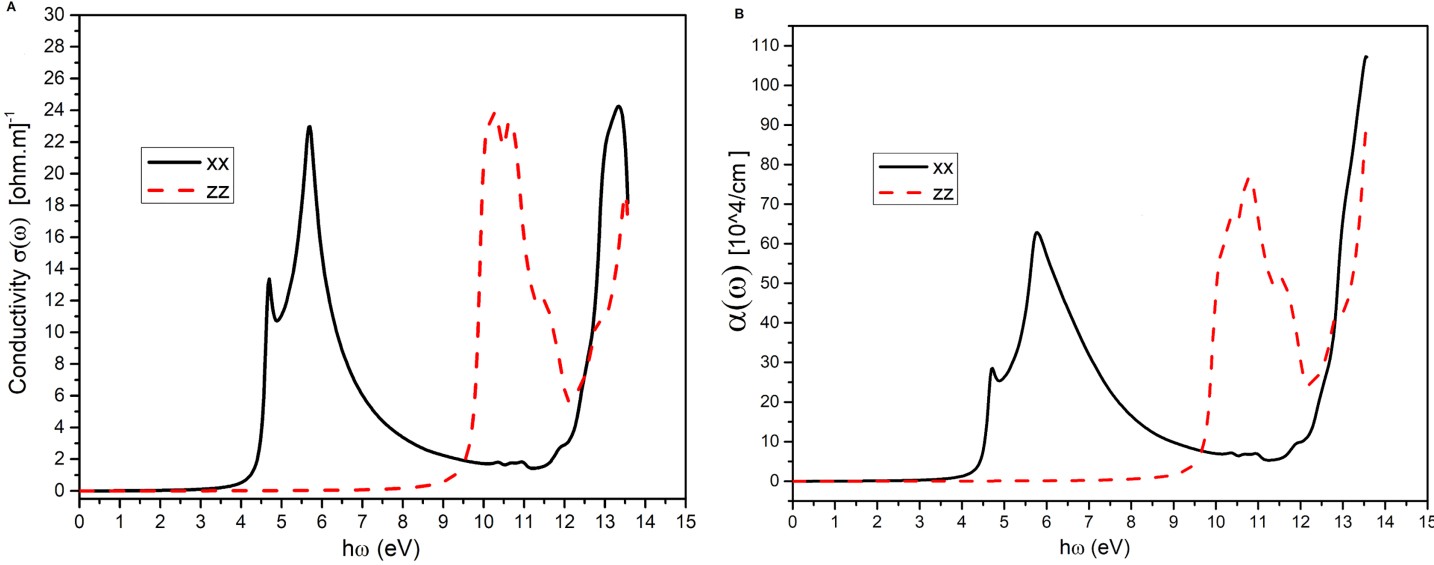

**Figure 2 The optical conductivity.** (A) The optical conductivities for ML h-BN in the xx in-plane and zz out of plane directions. (B) The absorption coefficients for ML h-BN in the xx in-plane and zz out of plane directions.

with the recent experimental evidence using optical spectroscopy of *Cassabois, Valvin & Gil (2016)* and the work of *Elias et al. (2019)* who used a combination of deep-ultraviolet photoluminescence and reflectance spectroscopy with atomic force microscopy, it is now confirmed that there is an indirect to direct bandgap crossover in h-BN bulk to the ML. The bandgap is much smaller than experiment in this work due to the well known problem of underestimation of GGA-DFT works.

## Dynamical optical conductivities and constants

The term "optical conductivity" means the electrical conductivity in the presence of an alternating electric field or dynamic electrical conductivity. The term "optical" here implies the entire frequency range, and is not restricted to just the visible region. The imaginary part of the dielectric constant is related to the real part of the AC conductivity and, therefore, to the optical reflectivity and transmittance (*Ashcroft & Mermin, 1976*). The real part of the dynamic electrical conductivity is connected with the energy absorbed by the electrons. The imaginary part of dielectric function, $\varepsilon_2(\omega)$, which represents absorption behavior, can be calculated from the electronic band structure. The real part of dielectric function, $\varepsilon_1(\omega)$, can be calculated according to Kramers-Kroing relation (*Kramers, 1927*; *de Kronig, 1926*) which represents the electronic polarization under incident light.

The optical conductivity and absorption behaviour of ML h-BN with respect to the photon energies are shown in Fig. 2. We see from the figure the maximum conductivity and absorption peaks are at ~ 6, 11 and 13.5 eV that is in the mid-UV and DUV regions considering the in-plane xx and perpendicular zz directions. There is also a small shoulder at around 4.5 eV. We also note that both the xx and zz directions have almost similar magnitudes for the conductivity and absorption coefficients $\alpha(\omega)$. The absorption coefficient peak value is in good agreement with the experimental work of *Li et al. (2012)*.

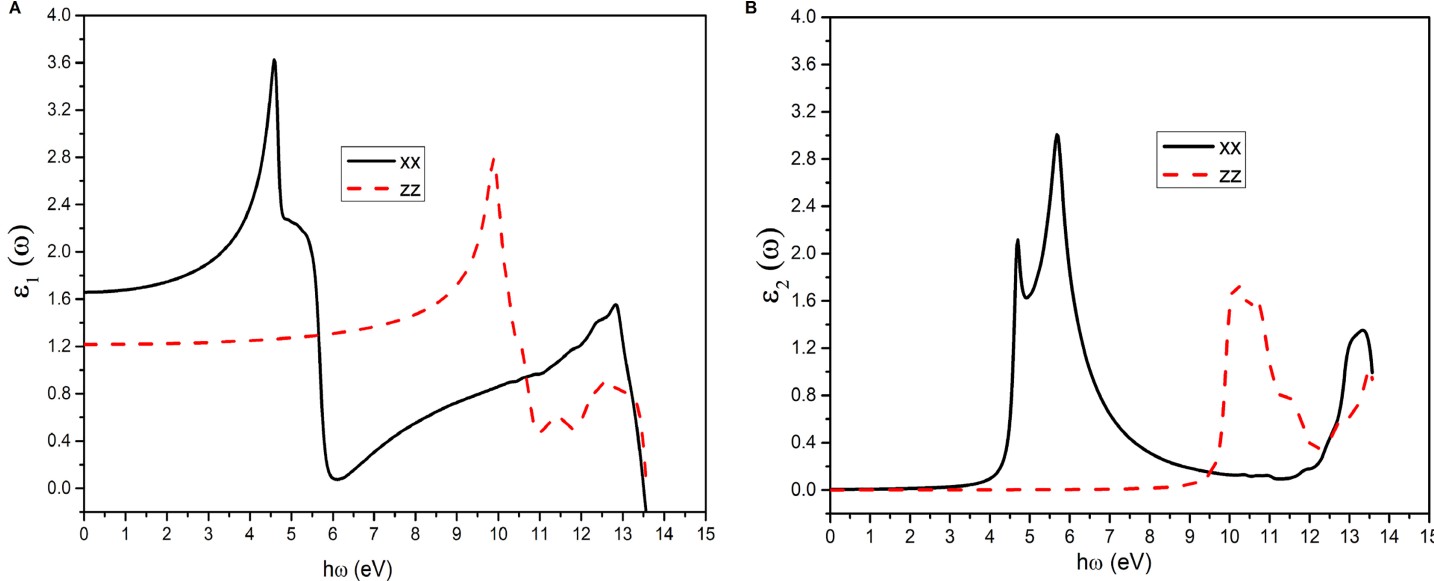

**Figure 3  The dielectric function.** (A) The real parts of the dielectric function in the xx and zz directions for ML h-BN. (B) The imaginary parts of the dielectric function in the xx and zz directions for ML h-BN.               

A comparison with the Figs. S3 and S6 shows that the ML h-BN retains the bulk characteristics for the conductivities and absorption coefficients α(ω), however, their magnitudes in the bulk are almost three times and two times more than the ML h-BN values respectively.

Figure 3 depicts the real and imaginary parts of the dielectric constant for the ML h-BN. The static values of the real dielectric constant are around 1.6 and 1.2 in the xx and zz directions respectively. Figures 3A and 3B curves show the anisotropic behaviour in the two directions as in the previous conduction and absorption spectra Fig. 2.

However, for bulk h-BN, a look at Fig. S4 shows that the static real dielectric constants are almost double in value around 3.6 and 2.3 in the in-plane and perpendicular directions respectively. A comparison of the magnitudes of the first sharp peaks of the imaginary dielectric constants Fig. 3B and Fig. S4B indicate the higher absorption capabilities of bulk as compared to that of ML h-BN.

From the values of $\varepsilon_1(\omega)$ and $\varepsilon_2(\omega)$ we can obtain the refractive index $n(\omega)$, reflectivity $R(\omega)$ and the extinction and absorption coefficients $k(\omega)$ and $\alpha(\omega)$ respectively from the relevant equations as detailed in the work (*Ramanathan & Khalifeh, 2021b*). The refractive index and extinction coefficients are shown in Fig. 4. Once again we notice the anisotropy in the xx and zz directions. The static refractive index values obtained are 1.3 and 1.1 in the in-plane and out of plane directions respectively and ML h-BN is transparent in the IR and visible region for xx direction and also in mid-DUV for the zz direction as evinced from the flatness of the curves. A direct comparison of the refractive indices of the bulk and ML is depicted in S5 and we see that the bulk values are consistently higher at 1.9 and 1.5 in the xx and zz directions respectively and reflect the higher electron density in the bulk. For the sake of better readability the static values of the refractive indices, the reflectivity and

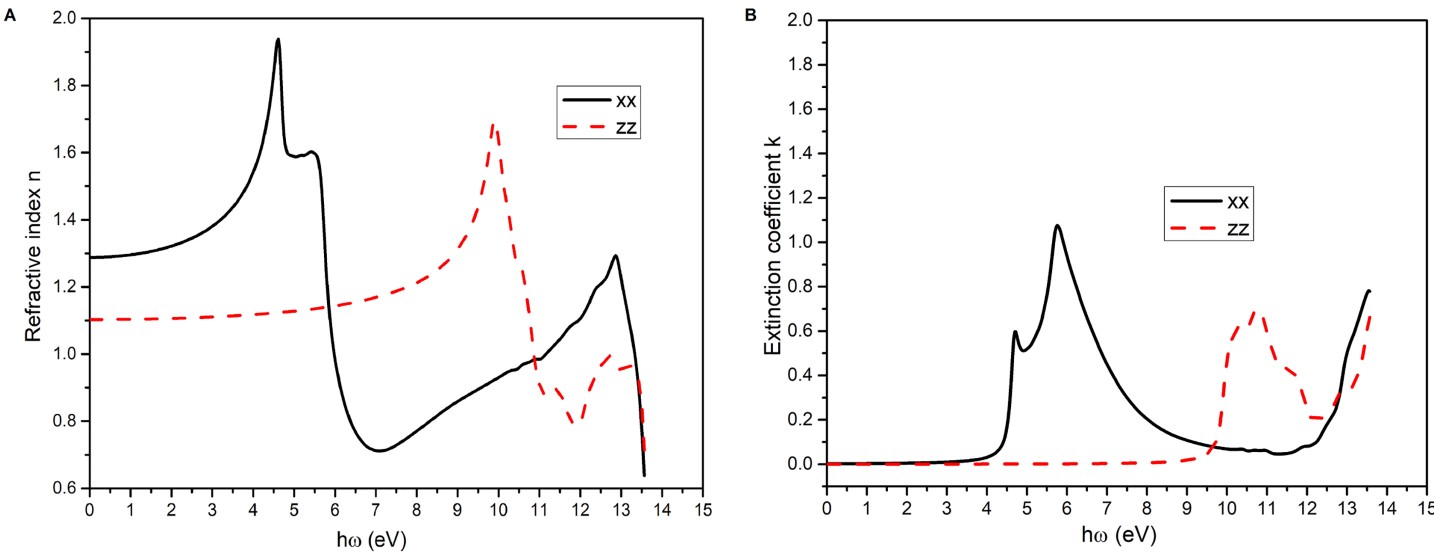

**Figure 4 The refractive index.** (A) The refractive indices in the xx and zz directions for ML h-BN. (B) The extinction coefficients in the xx and zz directions for ML h-BN.                         

**Table 2 Static values of optical properties.**

| Compound | Zeros of $\varepsilon_1(\omega)$ (eV) | | $\varepsilon_1(0)$ | | n(0) | | R(0) | | I(0) | |
|---|---|---|---|---|---|---|---|---|---|---|
| | **xx** | **zz** | **xx** | **zz** | **xx** | **zz** | **xx** | **zz** | **xx** | **zz** |
| hBN bulk | 5.84; 9.13 | 11.22; 12.20 | 3.64 | 2.32 | 1.91 | 1.52 | 0.097 | 0.043 | 0.0055 | 0.0024 |
| hBN ML | 13.51 | 13.56 | 1.66 | 1.22 | 1.29 | 1.10 | 0.016 | 0.002 | 0.0021 | 0.0004 |

**Note:**
The zero y-values of $\varepsilon_1(\omega)$ and the static optical constants (at $\omega$ = zero): dielectric constant $\varepsilon_1(0)$, refractive index n(0), reflectivity R(0) and absorption coefficient I(0) for the bulk and ML h-BN.

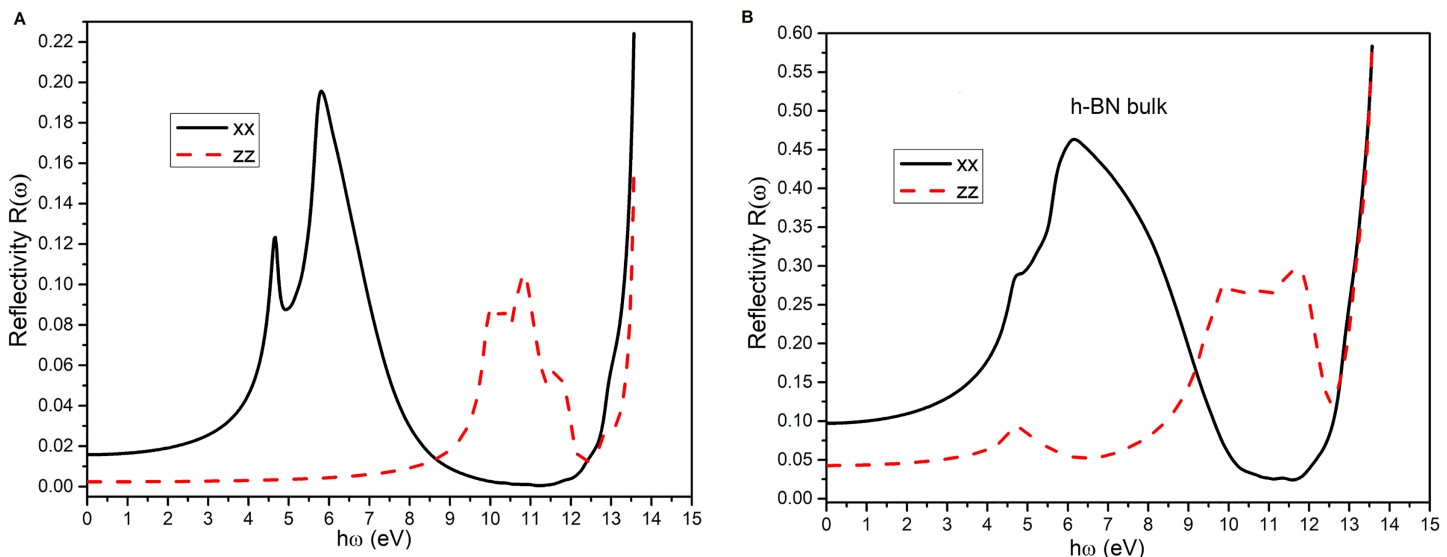

**Figure 5 The reflectivity of ML h-BN.** (A) The reflectivity plots for ML h-BN in the xx and zz directions. (B) The reflectivity plots for bulk h-BN in the xx and zz directions.                         

absorption coefficients for the bulk and ML h-BN in both the in-plane xx and out of plane zz directions have been gathered in Table 2. The table also includes the energies at the zero values of $\varepsilon_1(\omega)$.

The reflectivity plots for the ML and bulk h-BN are represented in Fig. 5. The static values of reflectivity are very low both for ML and bulk h-BN and continues to be low over a wide range of photon energies 0–9 eV for the perpendicular component which peaks in DUV region. Whereas, for the in-plane direction the reflectivity values are low in the visible and the first peak is around 6eV.

## DISCUSSION

The electronic structure calculations in this work confirms the previous literature results (*Cassabois, Valvin & Gil, 2016*; *Elias et al., 2019*; *Vuong et al., 2017*; *Schuster et al., 2018*) of the bandgap crossover of indirect to direct in h-BN bulk to ML. However, although the band gap values are in fair agreement with other theoretical works it is underestimated in comparison with experimental findings of 6.1 eV in ML h-BN (*Elias et al., 2019*) due to the shortcomings of the GGA exchange correlation potential. A more accurate calculation using the modified Becke-Johnson (mBJ) approximation (*Tran & Blaha, 2009*; *Ramanathan, 2021*) is expected to give a better estimate for the bandgap and is part of future work.

The optical properties are qualitatively consistent with other theoretical and experimental works (*Elias et al., 2019*; *de Kronig, 1926*; *Gao et al., 2009*; *Wang et al., 2016*; *Wang, Ma & Sun, 2017*) and reinstate the importance of h-BN in DUV optoelectronics. Essentially the h-BN ML mimics the behaviour of the bulk and the shape and peak positions for the optical conductivity and constants are similar but the magnitude is much lower owing to the higher values of the dielectric constants in the bulk. Since, the attractive wide band and optical properties in DUV is maintained in the ML h-BN, with the added advantage of a direct bandgap, ML h-BN is a promising candidate for nano-optoelectronics in the DUV region. Numerous applications as DUV photo detectors/emitters, possible integration with graphene optoelectronics due its flatness and similarity in structure and close lattice constant values, and in addition, its promise for realizing chip-scale DUV light sources are all waiting to be realized. Moreover, it can be argued that the optical properties of h-BN in the ultraviolet are very useful for the study of peptides and other biomolecules (*Elias et al., 2019*; *Ferreira et al., 2019*; *Henriques et al., 2020*).

## CONCLUSIONS

In conclusion, the study of the optoelectronic properties of ML h-BN using first principles FP-LAPW method with the GGA approximation shows an indirect K–M to direct K–K band transition when going from the bulk to the ML with a wide bandgap value of 4.65 eV, consistent with recent results. The excellent values of conductivity and absorption in the mid UV and DUV uphold the great promise of this material for use in DUV optics and energy conversions. A more accurate calculation using mBJ exchange correlation to reconfirm these very interesting optical properties is highly desirable and is the future direction.

## ACKNOWLEDGEMENTS

This work has been carried out at the theoretical physics laboratory, Department of Physics at the University of Jordan.

### Funding

The author received no funding for this work.

### Competing Interests

The author declares that she has no competing interests.

### Author Contributions

- Amall Ahmed Ramanathan conceived and designed the experiments, performed the experiments, analyzed the data, performed the computation work, prepared figures and/or tables, authored or reviewed drafts of the article, and approved the final draft.

### Data Availability

The raw data is available at figshare: Ramanathan, Amall (2022): Optical properties of BN bulk and monolayer. figshare. Dataset. https://doi.org/10.6084/m9.figshare.21174856.v1.

The Supplemental Figures are available in the Supplemental File and at figshare: Ramanathan, Amall (2021): Supplementary Information: 2D hexagonal boron nitride for solar energy conversions. figshare. Figure. https://doi.org/10.6084/m9.figshare.17708465.v2.

### Supplemental Information

Supplemental information for this article can be found online at http://dx.doi.org/10.7717/peerj-matsci.27#supplemental-information.

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
