# Peer review of "2D hexagonal boron nitride for solar energy conversions"

_PeerJ Materials Science, doi:10.7717/peerj-matsci.27_

## Round 0.1 · original submission · Major Revisions

The manuscript needs to be carefully revised in line with the concerns raised by the two independent experts selected for the review process.

Reviewer 1 ·

Basic reporting

no comment

Experimental design

no comment

Validity of the findings

Fine

Additional comments

The author have investigated the optical properties of the h-BN monolayer and the results of
the properties exhibited by the bulk and 2D lattice are systematically compared. The author
have performed the simulations utilizing the bulk and 2D h-BN monolayer whose lattice
constants are in agreement with the previous reports. Further, the electronic band structure of
the bulk and 2D lattice has also been reproduced. The outcomes of the work suggest that the
optical absorption of the h-BN occur in the deep UV region and therefore, they can be
utilized as the components in functional devices like optoelectronic and energy conversion. However, following points need to be taken care prior to its consideration for the publication.
1) The computational details for the simulation should be precisely mentioned. For
instance, the k-point sampling and the energy cutoff values should be given. Further,
the convergence test for the K-point and energy cutoff values should be carried out or
if already established and if considered from the previous report then please cite the
relevant work.
2) Improvement in the English more particularly the punctuations at appropriate
positions in the text is necessary.
3) The calculated values related to the optical properties like the static dielectric
constants, static refractive index for the bulk and the 2D h-BN lattices structure
should be consolidated in a table for the clearer comparative analysis.
4) The quality of the work will greatly be improved if the calculations would be
performed employing the HSE06 functional.

Reviewer 2 ·

Basic reporting

Main figures of this work only consist of simulated optical properties with respect to the incident light. Also, the author should also include better illustrations for readers' convenience, e.g. molecular structure of hBN.

Experimental design

No comment

Validity of the findings

What is the novelty of this work? Is this the first time performing DFT simulation to investigate optical property of hBN? Is full potential linearized augmented plane wave (FP-LAPW) method used by this work made any progress or discovered new findings?

Additional comments

No comment

---

## Round 0.2 · accepted · Accept

The revised version of the manuscript appears to have been modified and improved in accordance with the reviewers' comments.

Reviewer 1 ·

Basic reporting

The article follows scientific writing rules.

Experimental design

NA

Validity of the findings

Validated

Additional comments

I do agree with the author, HSE06 is time consuming.